# Working Conditions and Work Engagement by Gender and Digital Work Intensity

**Paula Rodríguez-Modroño** 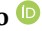

Department of Economics, Quantitative Methods and Economic History, Pablo de Olavide University, 41002 Seville, Spain; prodmod@upo.es

**Abstract:** Telework and other flexible working arrangements, which have exponentially expanded with new advancements in digitalization and the impact of COVID-19, are modifying working conditions and workers' engagement. Using the 'job demands-resources' model, we applied multivariate techniques to examine the different ways in which telework intensity impacts working conditions by gender. Increased intensity of remote working was positively associated with better skills and discretion and work engagement, while it was negatively associated with the other dimensions of job quality (particularly with working time quality). Even though women usually score higher than men in work intensity or working time quality, high intense female teleworkers experience a downturn with respect to these two items. Low and medium intensities of teleworking were positively associated with skills and discretion, working time quality, improved physical environment, and especially with better prospects and earnings. In conclusion, the intensity of teleworking and gender affect job quality and work engagement in different degrees, highlighting the importance of including these multiple effects on the design of flexible working arrangements.

**Keywords:** digital work; flexible working arrangements; telework; working conditions; gender studies





## 1. Introduction

In the last few decades, a growing number of tasks can be performed and surveilled anywhere and anytime thanks to digitalization, increasing flexibility within the labor market, and digital systems and technologies [1]. Technological development and greater global integration have led to an unprecedented rise of virtual work and global virtual teams, with its subsequent potential advantages and disadvantages [2]. This trend in increasing remote work and virtual work has intensified exponentially with the COVID-19 pandemic, where many individuals have been forced to telework from home due to government enforced lockdowns [3]. Many companies and organizations are now offering different flexible working arrangements (FWA) to their personnel and are planning a hybrid virtual model that combines remote work with time in the office for the post pandemic future.

Works in the available literature show that these FWA, such as different combinations of telework or virtual work, are completely modifying working conditions and job quality, in terms of job content; work intensity and job autonomy; working time arrangements and work–life balance; social environment, including interpersonal relationships at work and social support; and job insecurity and career development [4–7]. Therefore, with the rapid expansion of digital work, it is vital to monitor developments in the various dimensions of job quality and to assess if all remote workers experience the same impacts on working conditions and work engagement. Studies on the associations between virtual work and job quality are still scarce and inconclusive [8]. One possible explanation for these contradictory findings is that existing studies do not usually distinguish between different intensities of teleworking, neglecting that the impacts of telework may greatly differ depending on the type of flexible working arrangement [9–13]. Another one is the absence of a gender

lens in the analysis of digital work, despite technologies are not gender neutral and power relations in society determine the enjoyment of benefits from the use of them. Also, in the case of virtual work, this type of work is mainly carried out from home, with very different consequences for women and men due to the unequal distribution of unpaid care work between women and men [14,15].

Therefore, in this study we examine the relationship between teleworking [16,17] and several dimensions of job quality, incorporating two crucial axes of analysis (first, the differences by intensity of teleworking, and second, gender inequalities). To address these goals, the study is organized as follows. Next section presents the theoretical framework used to analyze the impacts of different intensities of teleworking on workers' job quality and engagement. Section 3 describes the materials and methods. Sections 4 and 5 outline the main results and discuss the findings, respectively. The article ends with the conclusions, limitations and directions for future research.

## 2. Literature Review

According to the 'job demands-resources' model (JD-R model), two different processes lead to work-related stress and motivation [18–20]. The first is the health-impairment process, in which exposure to adverse work demands (e.g., mental and physical workload) consume physical and psychological energy resources, leading to higher levels of exhaustion, which in turn are related to poorer health [21]. However, appropriate job resources may mitigate adverse effects of job demands. The second process is the motivation process, in which access to work resources contribute to stimulating positive job outcomes, such as higher levels of work engagement, which in turn are related to better health and well-being.

Telework modifies the existing balance between job resources and job demands. On the one hand, telework can create relevant job resources at both the interpersonal and job levels [19]. Telework offers more autonomy and flexibility, which usually leads to better work–life balance. Teleworkers report greater influence over how they organize their day and more overall hours to dispose of since they do not spend time commuting. The more the job autonomy that teleworkers have the greater the effort they put into their work [22,23]. Thus, employers gain from a more productive workforce which uses less space and is more cost effective, and workers gain from the prospect of a better work-life balance, thereby increasing levels of job satisfaction and organizational commitment. This situation may have a positive influence on employee well-being. Many advocates of telework note its benefits in promoting female labor force participation, due to the supposedly increased flexibility for accommodating productive and reproductive work [24–27].

On the other hand, telework can lead to an intensification of work, longer working hours, and the overlapping of work and home life, notably increasing stress and triggering health issues [28]. Teleworkers tend to work long and continuous hours and feel they must always be on call. The possibility to telework brings with it the risk of a blurring of boundaries between work and non-work life [29], increased work demands, the depersonalization of relationships at work, a lack of clarity in job roles and adverse effects on individual well-being. The achievement of work-life balance is more difficult where the borders between home and work are intentionally blurred as is the case for teleworkers [30]. Teleworkers face issues such as work intensification and excessive hours, the blurring of boundaries between paid work and private life, and social isolation [31]. Therefore, we expect a deterioration of working conditions in relation to working time, social and physical environment associated to increases in telework intensity.

This requirement of constant availability and instantaneous responsiveness, which characterizes many digital jobs and is likely to harm women more than men (as women are those who usually must juggle work with care, exacerbating gender inequalities) [32–35]. Literature on gendered impacts of telework and digitalization highlight the different implications for work–life balance of the flexibility associated with remote work [14,15,17,36]. Research shows that flexibility stigma associated to teleworking is gendered, and women making use of flexible work arrangements, especially mothers, were more likely to suffer

discrimination [24]. Therefore, our analysis incorporates the interactions between gender and intensity of telework as a crucial differentiating factor.

To explore changes in job quality, we use the seven indices constructed by Eurofound based on the JD-R model. We also include measures of work engagement since literature in the work organization field finds that better working conditions lead to higher levels of work-related outcomes, mostly measured in terms of lower job stress and higher work engagement [37,38]. We apply one-way and two-way ANOVA and OLS regression models to the analysis of all of these composite indices and scales to test the following hypotheses:

**Hypothesis 1 (H1).** *Telework intensity affects job quality dimensions and work engagement of workers.*

**Hypothesis 2 (H2).** *Gender and its interaction with telework intensity also impact on job quality dimensions and work engagement.*

### 3. Materials and Methods

The analyses are based upon cross-sectional data for 35 European countries, including the EU Member States, Norway, Switzerland, Albania, North Macedonia, Montenegro, Serbia and Turkey in 2015 from the last European Working Conditions Survey (EWCS). The EWCS is one of the very few comparative data sources that collects data on working conditions and workers' well-being [39]. Our database comprises 43,850 respondents for the 35 European countries.

Although telework is not directly addressed in the EWCS, this survey includes several questions based on the main place of work and the reported use of ICT, which allow us to create a proxy indicator that captures the incidence of telework in all EU Member States. We adapt the definition of ILO of telework: the worker uses ICT at least 1/2 of their work and their work is carried outside the employer's premises. We categorize teleworkers in three groups by intensity of telework: (a) high intensity of teleworking is defined as those workers out of the company premises at least several times a week and using ICT devices; (b) medium intensity of teleworking is defined as those workers using ICT devices and out of the company premises at least several times a month but less than several times a week; and (c) low intensity of teleworking is composed of those working in other locations other than the office less often and using ICT devices. The rest of the workers are included in the category 'not teleworking'.

Table 1 presents the descriptive statistics on the sample (all frequencies weighted). Approximately 12.7% of the workers present a high intensity of remote working, 4.5% are average teleworkers, 5.2% are occasional teleworkers and 77.6% do not do any telework. Women are less likely to perform telework with more intensity. Age is very similar but the number of children under 15 years-old is higher for teleworkers. Most high-intensity teleworkers have a tertiary education, are professional workers and many of them are freelancers and work in education or other services.

We include eight indices that comprise several dimensions of job quality and work engagement. To examine job quality, we use the framework proposed by Eurofound [19], which defines job quality as a multidimensional concept and distinguishes seven dimensions. Positive and negative features of the job are included, thus capturing the demands of the job but also the resources it provides to cope with demands [39]. Firstly, the physical environment index comprises 13 indicators related to specific physical hazards: posture-related (ergonomic) risks, ambient risks, and biological and chemical risks. Second, work intensity includes quantitative demands, time pressure, frequent disruptive interruptions, pace determinants, interdependency, and emotional demands. Third, the skills and discretion index measures the skills required in the job through 14 items that comprise the following components: cognitive dimension, decision latitude, worker participation in the organization, and training. Fourth, working time quality includes the incidence of long working hours, scope to take a break, atypical working times, working time arrangements, and flexibility. Fifth, the social environment index comprises 15 indicators to measure the extent to which workers experience (on the positive side) supportive social relation-

ships and (on the negative side) adverse social behavior such as bullying/harassment and violence at the workplace. The prospects index refers to employment status, career prospects, job security and downsizing. These job quality indices are measured on a scale from 0 to 100. Apart from work intensity, a higher index score would correspond to better job quality. Earnings refer just to one variable, namely monthly earnings.

**Table 1.** Descriptive statistics on the sample by intensity of teleworking (weighted) in 35 European countries (mean values or frequencies in %). Source: Own elaboration based on EWCS.

| Variables | Not TW | Low TW | Medium TW | High TW |
|:---:|:---:|:---:|:---:|:---:|
| **Total** | 34,042.9 | 2258.3 | 1976.1 | 5572.6 |
| | 77.63% | 5.15% | 4.51% | 12.71% |
| **Women** | 45.9% | 50.8% | 45.0% | 43.9% |
| **Age** | 46.93 | 44.91 | 48.04 | 46.39 |
| **Level of education** | | | | |
| Low (0–2) | 23.3% | 8.5% | 5.5% | 12.0% |
| Medium (3–4) | 51.2% | 45.2% | 39.4% | 39.0% |
| High (5–8) | 25.5% | 46.2% | 55.1% | 48.9% |
| **Children < 15** | 0.46 | 0.49 | 0.52 | 0.53 |
| **Workplace size** | | | | |
| 1 worker | 17.1% | 5.4% | 9.6% | 23.7% |
| 2–9 workers | 23.7% | 17.2% | 20.5% | 22.2% |
| 10–249 workers | 40.2% | 53.3% | 46.3% | 39.7% |
| 250+ workers | 16.2% | 22.4% | 21.9% | 12.3% |
| **Occupation** [1] | | | | |
| Managers | 4.6% | 7.1% | 9.1% | 8.9% |
| Professionals | 12.9% | 31.4% | 38.1% | 34.7% |
| Technicians & assoc. professionals | 12.9% | 18.3% | 19.4% | 13.1% |
| Clerical support | 9.5% | 18.8% | 11.5% | 13.7% |
| **Activity (NACE)** [1] | | | | |
| Industry | 17.6% | 17.5% | 16.1% | 9.3% |
| Commerce & hosp. | 20.7% | 15.1% | 13.0% | 14.2% |
| Public administration | 5.4% | 7.5% | 8.0% | 5.1% |
| Education | 5.8% | 11.8% | 16.3% | 17.6% |
| Health | 10.8% | 12.7% | 10.3% | 10.3% |
| Other services | 16.6% | 21.6% | 23.2% | 22.8% |

[1] This table only displays only ISCO and NACE categories with the highest percentage of workers.

Work engagement is defined as 'a positive, fulfilling, affective-motivational state of work-related well-being' [40], with favorable effects on workers' well-being, and job performance [41]. For the measurement of work engagement, we use the Utrecht Work Engagement Scale (UWES), based also on the JD-R model. The UWES, which is the most widely used operationalization of work engagement [42], originally consisted of a 17-item self-report questionnaire covering three dimensions of work engagement. The UWES has since been shortened, first to a 9-item version and later to a 3-item version [43,44], providing a simple measurement tool that can be easily incorporated into a variety of surveys. The three items added up in a single index measure: 'vigor' (bursting with energy), 'dedication' (enthusiasm with job), 'absorption' (time flies when working). The index score ranges from 0 to 100.

To test the theoretical hypotheses defined in the previous section, we compare differences in mean values and variances for the indices for each group of teleworkers using analysis of variance, including one-way and two-way ANOVA tests. First, we perform an analysis for each level of intensity of teleworking and, second, we introduce gender also as a factor, including its interaction with the different intensities of remote work. Afterwards, separate regression models are estimated for each dimension to examine the associations with telework intensity and gender. As occupational characteristics and personal circumstances could influence the relationships, the estimates were controlled for occupation,

economic activity, age, level of education, number of children under 15 years-old, and country of residence. We applied ordinary least square (OLS) regression models to the seven job quality dimensions and work engagement.

## 4. Results

### 4.1. Differences by Intensity of Telework

First, we compute one-way ANOVA to test if intensity of teleworking has a significant effect on job quality, and work engagement. There was a significant effect of intensity of teleworking at the $p < 0.05$ level for all indices: physical environment (F = 156.82, $p = 0.000$), work intensity (F test = 38.78, $p = 0.000$), skills and discretion (F test = 1112.89, $p = 0.000$), working time quality (F test = 149.65, $p = 0.000$), social environment (F test = 11.13, $p = 0.000$), prospects (F test = 149.06, $p = 0.000$), earnings (F test = 414.87, $p = 0.000$), and work engagement (F test = 59.38, $p = 0.000$). Hence, our analysis confirms Hypothesis 1 that there are significant differences by intensity of remote work.

Figure 1 shows that teleworkers score the maximum in skills and discretion (76.1), median monthly earnings (€2017.76), and work engagement (82.8). More intense teleworkers only outweigh the others in work engagement (79.5). Medium teleworkers (several times a month) score highest in physical environment (87.3), skills and discretion (68.1), prospects (69.2), and earnings (€1883). Those workers teleworking less often report better in working time quality (73.8) and social environment (78.2). Finally, those workers that never telework present the best work intensity (67).

Looking in depth at the divergences by telework intensity, high-intensity teleworkers (several times a week) report the worst working time quality and social environment, but they are the group with the highest earnings and work engagement. Workers with a medium intensity of teleworking (several times a month) exhibit the worst work intensity (quantitative demands, emotional demands and pace determinants and interdependency), although they score high in skill and discretion, prospects, earnings, and job satisfaction. Workers with a low intensity of teleworking have the best physical environment and working time quality. Lastly, those workers that never telework have the best work intensity, but their physical environment, skills and discretions, prospects, earnings, work engagement, and job satisfaction and are the lowest.

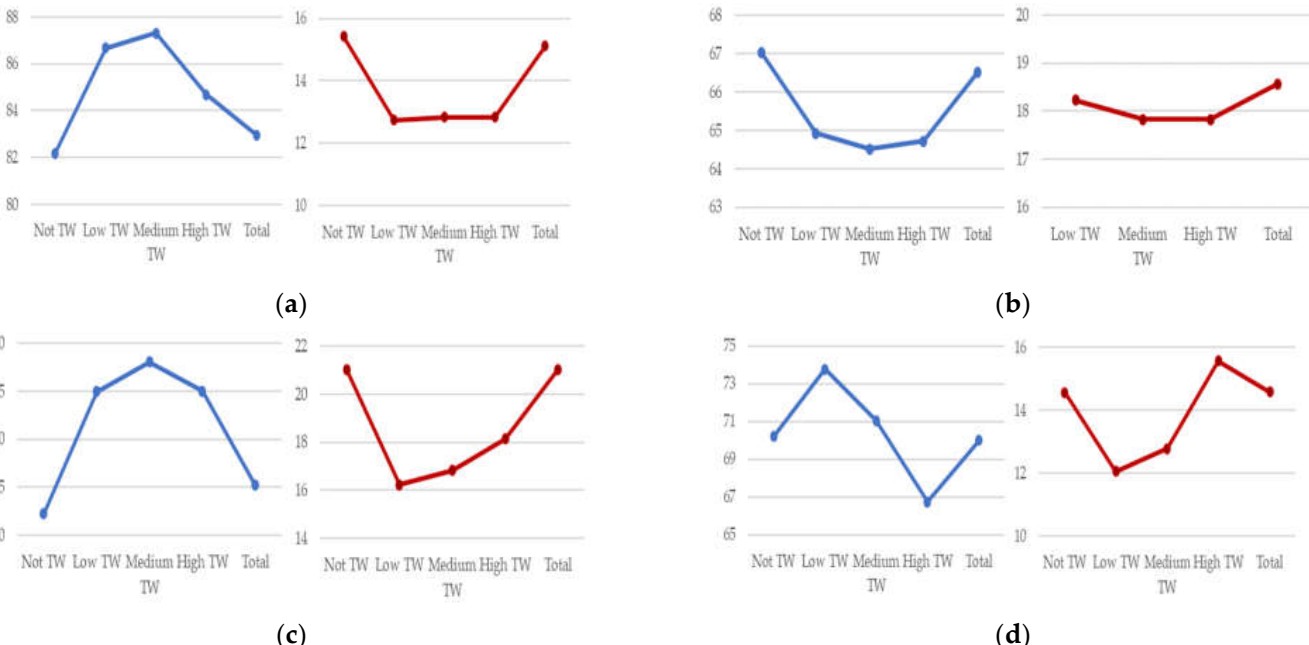

**Figure 1.** *Cont.*

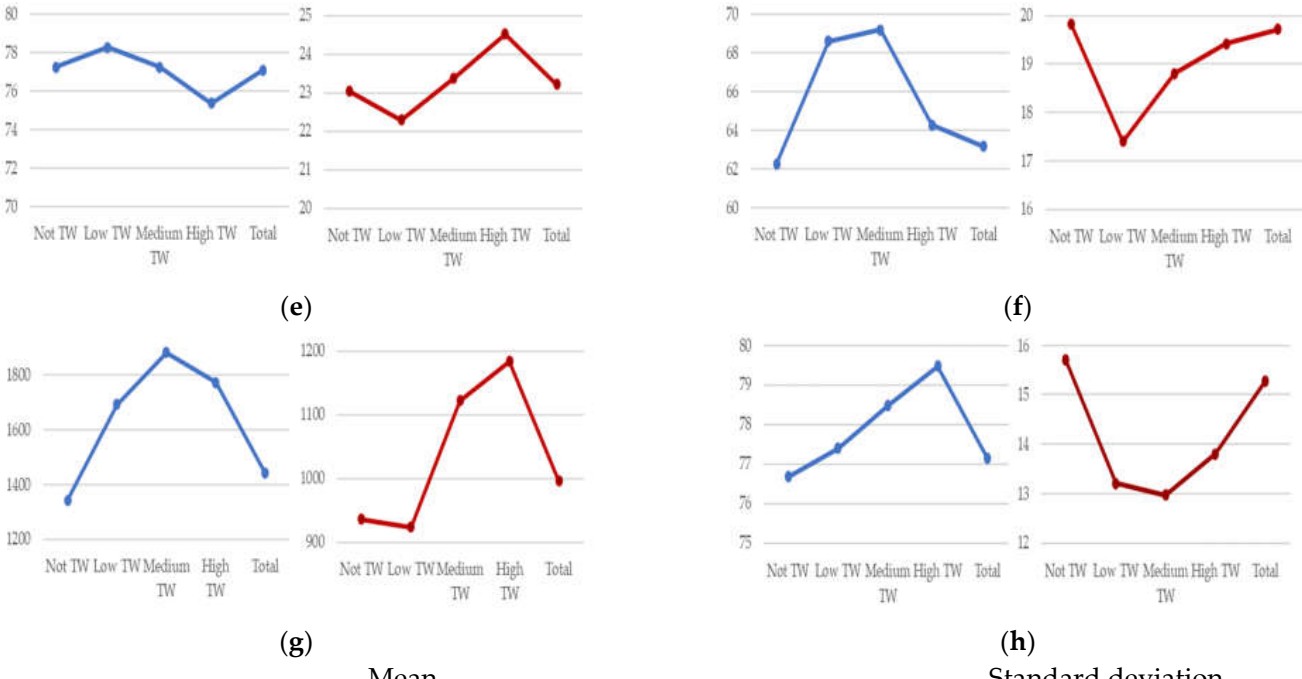

**Figure 1.** Mean and standard deviation of each work index, by intensity of teleworking, Europe 35 (weighted). Mean in blue and standard deviation in red. Source: Own elaboration based on 6th EWCS data. (**a**) JQI Physical Environment Index; (**b**) JQI Work Intensity (rev) Index; (**c**) JQI Skills and Discretion Index; (**d**) JQI Working Time Quality Index; (**e**) JQI Social Environment Index; (**f**) JQI Prospects Index; (**g**) JQI Earnings; (**h**) Work Engagement Index.

### 4.2. Differences by Intensity of Teleworking and Gender

A two-way ANOVA revealed that the effects of gender, intensity of telework and their interaction were also statistically significant for all of the indices or dimensions: physical environment (F = 323.44, $p$ = 0.000), work intensity (F test = 21.04, $p$ = 0.000), skills and discretion (F test = 487.19, $p$ = 0.000), working time quality (F test = 212.31, $p$ = 0.000), social environment (F test = 9.43, $p$ = 0.000), prospects (F test = 68.08, $p$ = 0.000), earnings (F test = 414.62, $p$ = 0.000), and work engagement (F test = 33.32, $p$ = 0.000).

Figure 2 summarizes the mean values for all indices by degree of intensity of telework and gender. Female workers report better results than male workers in physical environment, work intensity, working time quality and work engagement. Women present also slightly higher results in the skills and discretion index when they telework at least several times a month. Women who telework several times a month (medium teleworking) present the best physical environment, followed closely by women who telework less often. Female workers usually score higher than men in this variable since it is closely related to the sector or economic activity. Work intensity is better for non-teleworkers, with small differences by gender. On the contrary, skills and discretion increases with teleworking, reaching its highest for medium teleworkers: 68.5 and 67.7 for women and men, respectively. Working time quality is better for occasional teleworkers and women, 75.1 for low intense female teleworkers. While social environment, prospects and earnings are higher for male workers that telework occasionally. Work engagement is above average for female workers, particularly for high intense teleworkers (80.7).

Post hoc comparisons using the Tukey test (Appendix A), Bonferroni, Scheffe, and Sidak tests indicated that the mean scores for all intensities of teleworking were significantly different than for those not teleworking. Also, gender differences are significant except for the dimension of work intensity. The interaction of gender and frequency of teleworking is also significant for all dimensions, except for work intensity and prospects. All of these

results confirm Hypothesis 2, highlighting the relevance of gender in the analysis of the impacts of teleworking.

| | | JQI Physical environment | JQI Work intensity | JQI Skills & discretion | JQI Working time quality |
|---|---|---|---|---|---|
| **Not teleworking** | Women | 85.7 | 67.6 | 51.6 | 72.8 |
| | Men | 79.2 | 66.5 | 52.8 | 68.0 |
| **Low teleworking** | Women | 88.4 | 64.9 | 64.7 | 75.1 |
| | Men | 84.9 | 65.0 | 65.2 | 72.4 |
| **Medium teleworking** | Women | 89.1 | 64.5 | 68.5 | 73.2 |
| | Men | 85.9 | 64.5 | 67.7 | 69.3 |
| **High teleworking** | Women | 87.0 | 64.8 | 66.9 | 68.3 |
| | Men | 82.9 | 64.6 | 63.5 | 65.5 |

| | | JQI Social Environment | JQI Prospects | Earnings (median) | Work engagement |
|---|---|---|---|---|---|
| **Not teleworking** | Women | 77.1 | 61.9 | 974 | 77.1 |
| | Men | 77.4 | 62.6 | 1363 | 76.3 |
| **Low teleworking** | Women | 76.8 | 68.1 | 1374 | 78.2 |
| | Men | 79.8 | 69.1 | 1806 | 76.6 |
| **Medium teleworking** | Women | 76.9 | 67.7 | 1491 | 79.1 |
| | Men | 77.5 | 70.4 | 1864 | 77.9 |
| **High teleworking** | Women | 73.6 | 63.5 | 1447 | 80.7 |
| | Men | 76.7 | 64.9 | 1686 | 78.5 |

**Figure 2.** Mean values of each index, by intensity of teleworking and gender, Europe 35 (weighted). Source. Own elaboration based on 6th EWCS data.

### 4.3. Assessing the Effects of Each Telework Intensity

Following the ANOVA, separate OLS regression models were estimated for each indicator of job quality, levels of teleworking and gender, controlling for other variables that may influence in the results. Table 2 displays the standardized coefficients, standard errors and the level of significance. The variance inflation factor (VIF) of each regressor was less than 3.3 in all models, implying no multi-collinearity problems. We omit some other specifications we performed with other variables and the interactions among them, since the regression models did not improve.

Low intensity of telework had a positive effect on skills and discretion, working time quality, prospects, physical environment, and income, while it was negatively associated with work intensity. There were not statistically significant relations between low intensity of teleworking and social environment or work engagement. Medium intensity of teleworking was positively associated with skills and discretion, prospects, working time quality, income, and physical environment. But it was negatively associated with work intensity and social environment. A high intensity of teleworking had a strong positive relationship with skills and discretion, work engagement, prospects, and income. However, teleworking on a regular basis caused a decrease in working time quality, social environment, work intensity and physical environment.

Being a woman was positively associated with working time quality, physical environment and work intensity, but negative relations with skills and discretion, prospects, social environment, work engagement, and income. Education, particularly tertiary edu-

cation, presented positive relations with job quality and work engagement, except for the dimensions that captured work intensity and working time quality.

Finally, occupational categories and economic activities showed statistically significant associations with working conditions. Nevertheless, these OLS regressions and other models performed that distinguished by main occupational categories confirm that telework intensity plays a role in shaping organizational aspects of work regardless of the specific job or occupation under consideration. Being a manager was the occupation that increased in a larger extent skills and discretion, prospects, physical environment, work engagement, and social environment. Clerical workers performed also well in physical environment, better than professional and technicians, and better than the rest of occupations in terms of work time quality. Work intensity particularly worsened for managers and technicians.

**Table 2.** Standardized coefficients of OLS regressions in 35 European countries. Source: Own elaboration based on EWCS.

| | Physical Env | Work Intensity | Skills & Discretion | Working Time | Social Env | Prospects | Log Income | Work Engagement |
|---|---|---|---|---|---|---|---|---|
| Low teleworking | 1.144 *** | −0.905 ** | 5.189 *** | 3.237 *** | −0.282 | 3.208 *** | 0.129 *** | −0.277 |
| | (0.308) | (0.418) | (0.390) | (0.328) | (0.542) | (0.444) | (0.0157) | (0.343) |
| Medium teleworking | 0.882 *** | −2.235 *** | 6.683 *** | 1.153 *** | −0.972 * | 2.580 *** | 0.172 *** | −0.388 |
| | (0.323) | (0.439) | (0.409) | (0.344) | (0.572) | (0.465) | (0.0165) | (0.359) |
| High teleworking | −0.583 *** | −1.636 *** | 5.716 *** | −3.005 *** | −2.828 *** | 0.652 ** | 0.118 *** | 1.040 *** |
| | (0.200) | (0.272) | (0.254) | (0.213) | (0.365) | (0.289) | (0.0104) | (0.223) |
| Women | 3.525 *** | 0.471 *** | −4.097 *** | 3.815 *** | −1.286 *** | −1.829 *** | −0.346 *** | −0.416 *** |
| | (0.134) | (0.182) | (0.170) | (0.143) | (0.243) | (0.194) | (0.00696) | (0.149) |
| Age | 0.0775 *** | 0.233 *** | 0.109 *** | 0.106 *** | −0.0157 | −0.104 *** | 0.00337 *** | 0.0387 *** |
| | (0.00526) | (0.00714) | (0.00665) | (0.00560) | (0.00966) | (0.00762) | (0.000273) | (0.00584) |
| Children < 15 | −0.473 *** | −0.405 *** | 1.582 *** | −0.331 *** | −0.202 | 0.932 *** | 0.0735 *** | 0.564 *** |
| | (0.0804) | (0.109) | (0.102) | (0.0856) | (0.145) | (0.116) | (0.00409) | (0.0894) |
| Secondary education | 1.338 *** | −1.134 *** | 1.738 *** | −0.154 | −0.285 | 4.033 *** | 0.161 *** | 1.210 *** |
| | (0.181) | (0.246) | (0.229) | (0.193) | (0.338) | (0.262) | (0.00940) | (0.202) |
| Tertiary education | 3.183 *** | −3.275 *** | 7.956 *** | −0.0678 | 0.360 | 6.044 *** | 0.367 *** | 2.304 *** |
| | (0.222) | (0.301) | (0.281) | (0.236) | (0.408) | (0.320) | (0.0115) | (0.246) |
| NACE | 0.522 *** | 0.376 *** | 0.173 *** | 0.230 *** | −0.349 *** | −0.0264 | −0.000497 | 0.170 *** |
| | (0.0228) | (0.0310) | (0.0289) | (0.0243) | (0.0422) | (0.0330) | (0.00119) | (0.0254) |
| ISCO | −1.672 *** | 0.372 *** | −3.791 *** | 0.318 *** | −0.546 *** | −1.534 *** | −0.0689 *** | −0.816 *** |
| | (0.0324) | (0.0440) | (0.0410) | (0.0345) | (0.0589) | (0.0468) | (0.00168) | (0.0360) |
| Country | −0.053 *** | 0.016 ** | −0.025 *** | −0.097 *** | 0.055 *** | 0.00003 | −0.0091 *** | −0.102 *** |
| | (0.00621) | (0.00843) | (0.00786) | (0.00662) | (0.0113) | (0.00897) | (0.000320) | (0.00691) |
| Constant | 82.26 *** | 55.00 *** | 64.41 *** | 62.65 *** | 83.65 *** | 70.37 *** | 7.211 *** | 79.18 *** |
| | (0.416) | (0.564) | (0.526) | (0.442) | (0.758) | (0.600) | (0.0215) | (0.462) |
| Observations | 43,251 | 43,147 | 43,271 | 43,272 | 39,202 | 43,008 | 36,376 | 43,272 |
| R-squared | 0.178 | 0.042 | 0.348 | 0.046 | 0.007 | 0.079 | 0.217 | 0.040 |
| Prob > F | 0.0000 | 0.0000 | 0.0000 | 0.0000 | 0.0000 | 0.0000 | 0.0000 | 0.0000 |

Standard errors in parentheses *** $p < 0.01$, ** $p < 0.05$, * $p < 0.1$.

## 5. Discussion

The rise in teleworking opportunities with the COVID-19 saved many jobs and showed the potential of a digital workforce, but it also called attention to the positive and negative effects of teleworking on working conditions and work engagement. The prospect of an increased incidence of remote working post-crisis may be attractive for many workers but it also raises questions about working conditions and work engagement of workers and work-life balance, particularly for women. This study fills this gap in knowledge on the impacts of virtual work or telework, distinguishing the effects by intensity of teleworking [16,45]. Thus, providing a more nuanced understanding of the implications of teleworking on different dimensions of job quality. The present study is the first one to apply the job demands–resources model to examine groups of workers by both intensity of teleworking and gender, resulting in two main contributions.

First, our analysis confirms Hypothesis 1: teleworking intensity matters. It causes significant variations in job quality and work engagement. High telework intensity (several times a week) is positively associated with skills and discretion, prospects, income and work engagement. This result is consistent with recent studies which found a positive significant association for teleworking several days per week and high work engagement [16,20]. However, teleworking more frequently is negatively correlated with physical and social environment, work intensity and working time quality. This is also in line with other studies

showing that regular teleworking significantly limits the opportunities for supportive relationships in the workplace, reduces the likelihood of high well-being scores in the "workplace relationships" factor and impacts negatively on work–life balance [6,46]. All teleworking intensities, regardless of the frequency, are associated with higher autonomy and skills and discretion, functioning as a motivator or as a buffer to deal with job demands. On the contrary, work intensity is negatively related to all remote work, but only more frequent telework decreases working time quality.

Our second main contribution is that gender is a crucial axis of inequality in labor markets and society, and its effects interact with the intensity of teleworking, validating Hypothesis 2. Despite the fact that women usually score better than men in work intensity or working time quality, women who frequently telework experience a negative effect in these two items. These results are consistent with previous research findings on a distinct use of flexible working arrangements by men and women due to the unequal allocation of work and care by gender in European societies. This means that intense remote work may lead to a deterioration of work-life balance and job quality of workers, particularly women [17,23,24].

In fact, the COVID-19 pandemic has made visible that though telework has helped to protect workers, from completely exiting the labor force, there were negative consequences for teleworkers, particularly mothers. As schools and childcare facilities were closed, combining work from home with childcare and home schooling constituted an extraordinary burden for working parents, particularly women [47]. Hence, this crisis has been called "momcession", as one group stands out as faring especially poorly in labor force and unpaid work outcomes - working mothers with school-age or younger children [48]. Research suggests that mothers who teleworked were more likely to be interrupted during work hours [49], their productivity have suffered more [50] and faced a higher childcare burden than mothers who could not telework [51]. Although fathers' involvement increased a little, out of necessity rather than opportunity, mothers took on more of the additional unpaid care work [52]. Therefore, studies on the effects of remote work on workers' conditions and well-being during the pandemic find that mothers who teleworked during the pandemic were more likely to report feeling depressed, anxious, and lonely and an overall decline in work satisfaction [47,53].

## 6. Conclusions

Our research insights contribute to an understanding of the social and individual challenges that have resulted from the rise in remote working with digitalization and the platform economy, and especially since the outburst of the pandemic. Our study finds relevant associations between the intensity or frequency of teleworking, gender, and job quality and work-related outcomes through their impacts on job demands and resources. The main limitation from our research is the use of pre-pandemic data since COVID-19 has exponentially accelerated the extension of remote work and other FWA. We expect that there will be a surge of interest and demand for flexible working from workers and managers. There are also signs of change in managers' perception toward flexible working, and many companies plan to continue large-scale homeworking into the future [3]. Hence, our results highlight the importance of incorporating the analysis of these different associations and effects of telework intensity on working conditions in the design of FWA, labor market policies and other public policies. Further research with recent data is needed and should be focused on the spread of virtual work in a post-pandemic scenario with energy and transport crises.

In the first place, from the perspective of job quality and well-being we should be cautious about the use of teleworking on a permanent or exclusive basis. It would instead be advisable to use remote working occasionally but not exclusively. Also, organizational support to employees to minimize the negative effects of telework on employees' commitment is crucial [53].

In the second place, from a gender perspective, governments must take a gender-sensitive approach when promoting remote work. This policy approach should include greater public investments in early childhood education and care (ECEC) and paternity leave, as well as promoting and normalizing the use of telework across men and women. Expanding flexible working for all workers, not only for women, can help remove some of the existing stigma against flexible working, and the career penalty attached to it, by making it a norm rather than exception [54].

**Funding:** This research was funded by MCIN (Ministerio de Ciencia e Innovación)/AEI (Agencia Estatal de Investigació)/10.13039/501100011033/, grant number PID2019-105835RB-I00, and CENTRA, grant number PRY074/19.

**Institutional Review Board Statement:** Not applicable.

**Informed Consent Statement:** Not applicable.

**Data Availability Statement:** All microdata analyzed are from the European Working Conditions Survey, carried out by Eurofound. The Eurofound datasets are stored with the UK Data Service (UKDS) in Essex, UK, and are promoted online via their website https://www.eurofound.europa.eu/surveys/about-eurofound-surveys/data-availability#datasets (accessed on 15 June 2020).

**Conflicts of Interest:** The author declares no conflict of interest. The funders had no role in the design of the study; in the collection, analyses, or interpretation of data; in the writing of the manuscript, or in the decision to publish the results.

## Appendix A

**Table A1.** Two-way ANOVA results for intensity of teleworking, gender and their interaction. Source: Own elaboration based on 6th EWCS data.

| Source | Partial SS | df | MS | F | Prob>F |
|---|---|---|---|---|---|
| **Physical environment** | 491,731.63 | 7 | 70,247.376 | 323.44 | 0.0000 |
| itelework | 98,547.514 | 3 | 32,849.171 | 151.25 | 0.0000 |
| women | 63,271.077 | 1 | 63,271.077 | 291.32 | 0.0000 |
| itelework#women | 14,029.156 | 3 | 4676.3854 | 21.53 | 0.0000 |
| **Work intensity (rev)** | 50,611.245 | 7 | 7230.1778 | 21.04 | 0.0000 |
| itelework | 40,957.942 | 3 | 13,652.647 | 39.72 | 0.0000 |
| women | 359.50248 | 1 | 359.50248 | 1.05 | 0.3064 |
| itelework#women | 1956.9842 | 3 | 652.32807 | 1.9 | 0.1275 |
| **Skills & discretion** | 1,400,873.5 | 7 | 200,124.79 | 487.19 | 0.0000 |
| itelework | 1,394,570.6 | 3 | 464,856.88 | 1131.65 | 0.0000 |
| women | 1558.6158 | 1 | 1558.6158 | 3.79 | 0.0514 |
| itelework#women | 25,554.251 | 3 | 8518.0836 | 20.74 | 0.0000 |
| **Working time quality** | 305,623.29 | 7 | 43,660.47 | 212.31 | 0.0000 |
| itelework | 91,048.303 | 3 | 30,349.434 | 147.58 | 0.0000 |
| women | 42,756.193 | 1 | 42,756.193 | 207.92 | 0.0000 |
| itelework#women | 6270.5508 | 3 | 2090.1836 | 10.16 | 0.0000 |
| **Social environment** | 35,506.667 | 7 | 5072.381 | 9.43 | 0.0000 |
| itelework | 21,087.961 | 3 | 7029.3204 | 13.07 | 0.0000 |
| women | 9903.8665 | 1 | 9903.8665 | 18.41 | 0.0000 |
| itelework#women | 11,063.559 | 3 | 3687.8531 | 3687.853 | 0.0001 |
| **Prospects** | 183,337.05 | 7 | 26,191.007 | 68.08 | 0.0000 |
| itelework | 169,027.04 | 3 | 56,342.346 | 146.46 | 0.0000 |
| women | 7070.9896 | 1 | 7070.9896 | 18.38 | 0.0000 |
| itelework#women | 2073.0735 | 3 | 691.02449 | 1.8 | 0.1455 |
| **Earnings (logincome)** | 1370.6135 | 7 | 195.80,193 | 414.62 | 0.0000 |
| itelework | 623.84362 | 3 | 207.94787 | 440.34 | 0.0000 |
| women | 196.4288 | 1 | 196.4288 | 415.95 | 0.0000 |
| itelework#women | 7.4868486 | 3 | 2.4956162 | 5.28 | 0.0012 |
| **Work engagement** | 134.16118 | 7 | 19.165883 | 27.84 | 0.0000 |
| itelework | 101.98541 | 3 | 33.995136 | 49.38 | 0.0000 |
| women | 25.868528 | 1 | 25.868528 | 37.58 | 0.0000 |
| itelework#women | 11.82789 | 3 | 3.94263 | 5.73 | 0.0006 |

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
