# Peer review of "Working Conditions and Work Engagement by Gender and Digital Work Intensity"

_information, doi:10.3390/info13060277_

Round 1

Reviewer 1 Report

The manuscript presents an analysis of the effect of work intensity/frequency of telework on various aspects of the work experience of a sample of workers in Europe, with specific attention to gender differences in the outcomes observed.    It makes the useful point that not all telework experiences are the same, so attempting to distinguish among different types of telework is an important aspect of a more nuanced analysis of the phenomenon.

The one major concern I have with the study is that I didn’t see an attempt in the analysis to control for type of occupation.  The description of the data set indicates that telework of all levels of intensity is significantly more common among professionals.  It is possible that some of the observed effects, then, are the result of the fact that teleworkers are likely to be professionals, not of teleworking itself.  If possible, the author should include occupation in the analysis to determine if the effects of telework vary by occupational type.  This should be possible, given that occupation type is identified in the data set described.

The study also uses a single, global measure of job satisfaction.  This is a fairly weak measure, since it’s more typical to use multiple measures and/or to use factor-specific measures of job satisfaction.  Does the data set contain additional measures of job satisfaction that could be included?  If so, this would provide a more persuasive analysis

Finally, there is little discussion in the article of the relevant literature on women’s experience of telework.  Much has been written about how working remotely affects women’s domestic responsibilities (not just child care, but domestic labor of various sorts).  During the Covid pandemic, there has been much comment on how telework enabled women to stay connected to the labor market but also tended to intensify traditional gender roles at home, particularly when parents were required to oversee children’s schooling at home.  The article is not overly long, so would benefit from a review of some of this literature and a more detailed effort to identify any conclusions in that literature that should be modified in light of the study’s findings.

Reviewer 2 Report

The paper examines working conditions and workers ’engagement as a result of new advancements in digitalization and the impact of Covid-19.

The paper presents the objectives and working hypotheses, the research methodology used and the statistical tools. The results are clearly described.

I appreciate that the discussion section should be extended by comparison with other recent results in this field.

Also at the end of the paper there should be a separate paragraph of conclusions.

Minor editing error are – line 108 - and (c) ) low intensity; line 126 - demands. and distinguishes.

Overall, the paper has a structure suitable for research.

The paper is suitable for publication after some minor corrections.

Reviewer 3 Report

Thank you for the opportunity to revise the paper: “Working conditions and work engagement by gender and intensity of digital work”

First of all, I think the paper deal with an important topic and it is generally well written. I have, however, a few suggestions that I hope the author finds useful:

I don´t understand well why you suggest the null hypotheses. On the contrary, most papers propose relationships in which they expect a significant result to happen.

Related to the previous point, I would expect the paper to include more arguments why the proposed relationships may happen. So far there is a nice introduction to the topic, but there aren´t many arguments/theories pointing towards what can be expected to find (even more in the case of the gender hypothesis).

About remote work and digital work, I think the references of Jiménez et al. (2017) and maybe Taras et al. (2019) would be helpful:

Jimenez, A., Boehe, D. M., Taras, V., & Caprar, D. V. (2017). Working across boundaries: Current and future perspectives on global virtual teams. Journal of International Management, 23(4), 341-349.

Taras, V., Baack, D., Caprar, D., Dow, D., Froese, F., Jimenez, A., & Magnusson, P. (2019). Diverse effects of diversity: Disaggregating effects of diversity in global virtual teams. Journal of International Management, 25(4), 100689.

Could common method bias affect your results?

In the discussion, it would be convenient to include a paragraph with the limitations of the study, and the potential future areas of research that arise from your paper for interested researchers.

Finally, I see the abstract in a “structured” way. This is of course OK if this is what the journal requires, but please check this is the case.

Good luck!

Round 2

Reviewer 1 Report

The revision addresses the most significant comments made in my initial report.  I would still like to see, in the discussion, an effort more clearly to indicate the relative significance of occupation vs. intensity of remote work in shaping the results the author found (are both equally significant?  did introducing occupation MODIFY the initial reports resulted).  Occupation (and other controls) have been added to the analysis, but the discussion is still a little hazy on what the results of that addition were.

Reviewer 3 Report

The paper has improved but please note that VIFs are used to check if there is multicollinearity, not common method bias. For common method bias, at the very least, you have to check Harman´s factor (or other more sophisticated methods).
